# EnvC Homolog Encoded by *Xanthomonas citri* subsp. *citri* Is Necessary for Cell Division and Virulence

**DOI:** 10.3390/microorganisms12040691

**Published:** 2024-03-29

**Authors:** Michelle M. Pena, Thaisa Z. Martins, Doron Teper, Caio Zamuner, Helen A. Alves, Henrique Ferreira, Nian Wang, Maria Inês T. Ferro, Jesus A. Ferro

**Affiliations:** 1Agricultural and Livestock Microbiology Graduation Program, School of Agricultural and Veterinarian Sciences, São Paulo State University (UNESP), Jaboticabal 14884-900, SP, Brazil; mpmaclellan@uga.edu (M.M.P.); thaisazmartins@gmail.com (T.Z.M.); 2Department of Plant Pathology and Weed Research, Institute of Plant Protection Agricultural Research Organization (ARO), Volcani Institute, Rishon LeZion 7505101, Israel; doront@volcani.agri.gov.il; 3Biochemistry Building, Institute of Biosciences, São Paulo State University (UNESP), Rio Claro 13506-900, SP, Brazil; c.zamuner@unesp.br (C.Z.); henrique.ferreira@unesp.br (H.F.); 4Department of Agricultural, Livestock and Environmental Biotechnology, School of Agricultural and Veterinary Sciences, São Paulo State University (UNESP), Jaboticabal 14884-900, SP, Brazil; helen.penha@unesp.br (H.A.A.); maria.ferro@unesp.br (M.I.T.F.); 5Citrus Research and Education Center, Department of Microbiology and Cell Science, Institute of Food and Agricultural Sciences, University of Florida, Lake Alfred, FL 33850, USA; nianwang@ufl.edu

**Keywords:** citrus canker, peptidoglycan hydrolase, periplasmic protein, cell division

## Abstract

Peptidoglycan hydrolases are enzymes responsible for breaking the peptidoglycan present in the bacterial cell wall, facilitating cell growth, cell division and peptidoglycan turnover. *Xanthomonas citri* subsp. *citri* (*X. citri*), the causal agent of citrus canker, encodes an *Escherichia coli* M23 peptidase EnvC homolog. EnvC is a LytM factor essential for cleaving the septal peptidoglycan, thereby facilitating the separation of daughter cells. In this study, the investigation focused on EnvC contribution to the virulence and cell separation of *X. citri*. It was observed that disruption of the *X. citri envC* gene (Δ*envC*) led to a reduction in virulence. Upon inoculation into leaves of Rangpur lime (*Citrus limonia* Osbeck), the *X. citri* Δ*envC* exhibited a delayed onset of citrus canker symptoms compared with the wild-type *X. citri*. Mutant complementation restored the wild-type phenotype. Sub-cellular localization confirmed that *X. citri* EnvC is a periplasmic protein. Moreover, the *X. citri* Δ*envC* mutant exhibited elongated cells, indicating a defect in cell division. These findings support the role of EnvC in the regulation of cell wall organization, cell division, and they clarify the role of this peptidase in *X. citri* virulence.

## 1. Introduction

*Xanthomonas citri* subsp. *citri* (*X. citri*) is a phytopathogenic Gram-negative bacterium and the causal agent of citrus canker, a severe disease, which affects all economically important citrus varieties worldwide, causing significant economic losses [1]. Gram-negative bacteria feature a dense peptidoglycan (PG) layer situated in the periplasmic space, positioned between their outer and inner membranes [2]. This polymer is composed of glycan strands of β-1,4-glycosidic bond-linked N-acetylglucosamine (GlcNAc) and N-acetylmuramic acid (MurNAc) disaccharides, cross-linked by short peptides, playing a crucial role in upholding and sustaining the shape and integrity of the cell [3]. Two main classes of peptidoglycan-lytic enzymes are responsible for the PG’s assembly: the glycosidases cleaving the glycan backbone and the amidases (or peptidases) cleaving the peptide side chain [4].

The *X. citri* strain 306 (GenBank accession number: AE008923.1) possesses nine proteins sharing the M23 peptidase domain, with four of them being hypothetical proteins (Appendix A). One of these proteins—XAC0024 (GenBank accession number: AAM34916.1)—is a homolog of *Escherichia coli* (*E*. *coli*) EnvC (GenBank accession number: EGO4467787.1), a widely distributed protein in bacteria [5]. EnvC is highly conserved among Gram-negative bacteria and functions as part of the septal ring apparatus [6]. In *E. coli*, EnvC is a periplasmic peptidase, which plays a role in septal peptidoglycan splitting and daughter-cell separation [7]. Deletion of the *E. coli envC* gene, like other genes encoding LytM domain hydrolases, such as *nlpD*, *ygeR* and *uebA*, leads to the formation of long cell chains, suggesting a defect in cell separation [7]. It was demonstrated that EnvC controls cell separation by activating PG-degrading amidases AmiA and AmiB [8]. Homologs of *envC* have been shown to perform similar functions in various bacterial species and are essential for the pathogenicity of several animal bacterial pathogens, including *Vibrio cholera*, enterohemorrhagic *E. coli* and *Fusobacterium nucleatum* [9,10,11]. The *envC* homolog of *Pseudomonas aeruginosa* was identified to be functionally redundant to *nlpD*, as deletion of both led to the formation of long cell chains and enhanced sensitivity to high temperature and antimicrobial compounds [12]. 

The role of LytM factors and PG amidases in pathogenicity and cell division was recently demonstrated in *X. campestris* pv. *campestris* [13]. Deficiency in cell separation was observed in either *nlpD* or *envC* deletion strains; however, deletion of the single gene *nlpD* had a significant effect on virulence and induction of hypersensitive response in non-host plants, while deletion of *envC* did not significantly affect host interactions [13]. 

In the present study, the *envC* homolog (GenBank XAC0024) of *X. citri* strain 306 was characterized. It was observed that *envC* was essential for virulence but did not completely compromise the ability of *X. citri* to trigger weak symptoms in a susceptible host genotype. Additionally, *X. citri* Δ*envC* displayed an altered cell shape compared with the wild-type (wt) strain. Similar to observations in other Gram-negative bacteria, the *X. citri envC* gene seems to play a role in daughter-cell separation. Moreover, the sub-cellular localization of *X. citri* EnvC protein fused with the mCherry fluorophore (EnvC-mCherry) is consistent with the protein occupying the periplasmic region. 

## 2. Materials and Methods

### 2.1. Bacterial Strains, Plasmids and Culture Condition

The bacterial strains and plasmids utilized in this study are detailed in Appendix A [14,15,16]. *E. coli* strains DH10B, SM10ʎpir and HST08–Stellar used for cloning were cultivated at 37 °C in a LB/LB-agar medium [17]. Growth in the liquid medium was conducted at 250 rpm (shaker) for 14–16 h, while growth in the solid medium took place in a bacteriological incubator for 14–16 h. *X. citri* 306 strains were cultivated at 30 °C in a NYG-rich medium (3 g/L yeast extract, 5 g/L peptone, 20 g/L glycerol, pH 7.0), nutrient broth (NB) medium (3 g/L meat extract, 5 g/L peptone) or on NB-agar plates (NB medium containing 15 g/L agar) supplemented with L-arabinose (0.05% *w*/*v*), starch (0.2% *w*/*v*) and sucrose (5% *w*/*v*) when required. *X*. *citri* was grown for 48 h either in liquid NYG-rich medium at 250 rpm (shaker) or in solid NB medium in a bacteriological incubator. The antibiotics carbenicillin and kanamycin or ampicillin and spectinomycin were used when required at a concentration of 50 μg/mL and 100 μg/mL, respectively.

### 2.2. Mutant Construction

#### 2.2.1. Partial Deletion of XAC0024 Nucleotide Sequence

The *X. citri* strain containing the disrupted XAC0024 gene (mutant Δ*envC*) was obtained through site-directed mutagenesis using the overlap extension approach in the polymerase chain reaction [18]. To generate the XAC0024 mutant, *X. citri* genomic DNA served as a template in a two-step PCR process. In the initial step, primers A(F) and B(R), as well as C(F) and D(R) (Appendix A), were employed in separate reactions to generate fragments AB and CD, respectively. Subsequently, these fragments were fused through a double-joint PCR. The PCR reaction, each with a final volume of 20 µL, contained 26.5 ng of DNA, 0.2 mM of each dNTP, 1U of Phusion High Fidelity DNA Polymerase (New England Biolabs, Ipswich, MA, USA), 0.5 μM of each primer and 3% of DMSO for primer pairs A–B and 5% of DMSO for primer pairs C–D. The PCR conditions comprised an initial denaturation at 98 °C for 30 s, followed by 35 cycles of denaturation at 98 °C for 10 s, annealing at 69 °C for 30 s and extension at 72 °C for 30 s. The reaction concluded with a final extension at 72 °C for 10 min, executed in a Veriti^®^ 96-Well Thermal Cycler (Applied Biosystems, Waltham, MA, USA). The resulting PCR products (fragments AB and CD) underwent purification using the Wizard^®^ SV Gel and PCR Clean-Up System (Promega, Madison, WI, USA), and they were quantified using the NanoDrop^®^ ND-1000 spectrophotometer (Thermo Fisher Scientific, Waltham, MA, USA). The size of the amplified fragments AB and CD was confirmed via agarose gel electrophoresis (Appendix A). The double-joint PCR step was performed using fragments AB and CD as templates and the primers A(F) and D(R) to generate the AD fragment, using 3% DMSO and the same PCR conditions described above, except for a final volume adjustment to 50 µL. The resulting product was subjected to 0.7% agarose gel electrophoresis (Appendix A), and the band with the expected size was recovered from the gel using the Wizard^®^ SV Gel and PCR Clean-Up System (Promega). The purified DNA fragment was quantitated in a NanoDrop^®^ ND-1000 spectrophotometer (Thermo Fisher Scientific) and submitted to a PCR reaction to add a 3′-A overhang to the ends of the AD fragment. This reaction utilized a final concentration of 0.2 mM dATP and 1 U of Platinum^®^ Taq DNA Polymerase Recombinant (Invitrogen, Waltham, MA, USA) in a 50 µL reaction for 10 min at 72 °C in a GeneAmp^®^ PCR System 9700 (Applied Biosystems).

#### 2.2.2. Deletion Vector Construction

The PCR-amplified AD fragment, featuring 3′-A overhangs, was ligated into the pGEM^®^-T Easy plasmid (Promega) using T4 DNA ligase, according to the manufacturer’s instructions. The ligation reaction comprised 50 ng of plasmid and 118 ng of insert in a final volume of 10 µL. Subsequently, 2 µL of the ligation reaction was used to transform 50 µL of chemically competent *E. coli* DH10B cells (Invitrogen), following the protocol described in Ref [17]. Recombinant bacteria carrying the plasmid DNA harboring the AD fragment were selected by plating them onto agar plates containing solid LB medium, 100 µg/mL of ampicillin, 0.1 mM of Isopropyl-β-D-1-thiolgalactopyranoside (IPTG) and 0.0032% of 5-bromo-4-chloro-3-indolyl-β-D-galactoside (X-gal). After a 16 h incubation period at 37 °C, two white colonies were picked, and their recombinant plasmid was isolated using the Wizard^®^ Plus SV Minipreps DNA Purification System (Promega), following the manufacturer’s instructions. The presence of the AD fragment was checked via PCR using the vector primers M13/pUC F-20 and M13/pUC R-48 (Appendix A), with subsequent confirmation of the sequence via sequencing on an ABI 3730xl DNA analyzer (Applied Biosystems) utilizing the same vector primers. Next, the recombinant plasmid was digested with *ApaI* and *SalI* restriction enzymes (New England Biolabs), and the AD fragment was recovered from agarose gel, as previously described. This fragment was then ligated into the suicide pOK1 plasmid previously digested with the same enzymes. The ligation reaction was used to transform chemically competent *E. coli* SM10 λpir cells, following the protocol described in Ref [17]. The recombinant bacteria carrying the recombinant pOK1 plasmid were selected by plating them onto agar plates containing solid LB medium and 100 µg/mL of spectinomycin. The purification of pOK1 plasmid DNA was accomplished using the Wizard^®^ Plus SV Minipreps DNA Purification System (Promega), and the presence of the AD fragment was confirmed by PCR using A(F) and D(R) primers.

#### 2.2.3. Mutant Generation

The pOK1 suicide vector, containing the A–D sequence (Appendix A), was employed to excise bases 370–962 of the *X*. *citri* XAC0024 gene (1236 bp). This deletion was achieved by integrating the suicide vector into chromosomal DNA through double-cross-over homologous recombination. Electrocompetent *X. citri* 306 wild-type cells were transformed with the recombinant pOK1 vector, as described in Ref. [19]. The screening of the deletion mutant followed the methodology described in Ref. [20], using NB-agar medium with the addition of spectinomycin antibiotic. Since the pOK1 vector has the *SacB* gene, a positive selection for loss of the vector was achieved via growth on sucrose. Colonies, which grew in sucrose but not in the presence of spectinomycin, were selected, and their genomic DNA extraction was carried out using the Wizard^®^ Genomic DNA Purification Kit (Promega), following the manufacturer’s instructions. The deletion was confirmed by PCR reaction using 50 ng of mutant and *X. citri* 306 wt genomic DNAs, GoTaq^®^ DNA Polymerase, as well as primers A(F) and D(R). The amplicons were visualized in 1% agarose gel, and amplicons with the expected length were then sequenced, and the confirmed mutant was named *X. citri* Δ*envC* (Δ*envC*).

### 2.3. Mutant Complementation

For complementation of the Δ*envC* mutant, a 2236 bp fragment encompassing the genome region in bases 25382–27617 was amplified. This fragment contained 1236 bp of the ORF XAC0024, along with a region spanning 500 base pairs upstream of the 5′ end and 500 base pairs downstream of the 3′ end. The amplification was carried out using the primers 0024_500_IF_F/0024_500_IF_R (Appendix A) and Q5 High-Fidelity DNA Polymerase (New England Biolabs). Subsequently, the PCR-amplified fragment was ligated into the *Xho*I site of the pMAJIIc plasmid [17], employing the In-Fusion HD Cloning Kit (Takara Bio USA, Inc., San Jose, CA, USA) according to the manufacturer’s recommendations. The complementation plasmid was confirmed by DNA sequencing (primer XAC0024_mcherry_F; Appendix A) and was used to transform the *X. citri* Δ*envC* strain, resulting in production of the *X. citri* Δ*envC* pMAJIIc-*envC* (*X. citri* Δ*envC amy*:pMAJIIc-*envC*) strain. 

### 2.4. Sub-Cellular Localization

#### 2.4.1. Vector Construction

For construction of the pMAJIIc-*envC* plasmid, enabling the sub-cellular localization of proteins fused with mCherry fluorescent protein, the *X. citri envC* gene (XAC0024) sequence was amplified via PCR with primers 0024_IF_F and 0024_IF_R (Appendix A) using *X. citri* 306 genomic DNA as a template, and the resulting product was ligated into the *Xho*I site of the pMAJIIc plasmid [17] using the *In-Fusion HD Cloning Kit* (Takara Bio USA, Inc.), following the manufacturer’s guidelines. The plasmid construction was confirmed by DNA sequencing (primers pGCD21-F and XAC0024_mcherry-F; Appendix A) and was used to transform electrocompetent *X. citri* 306 strain cells, generating the *X. citri* pMAJIIc-*envC* (*X.citri amy*:pMAJIIc-*envC*) strain.

#### 2.4.2. Fluorescence Microscopy 

The initial cultures of *X. citri* wt and *X. citri* pMAJIIc-*envC* were prepared by cultivating bacteria in 5.0 mL of NB medium for approximately 16 h at 30 °C and 200 rpm. The cultures were then diluted to an OD 600 nm of 0.1 using fresh NB medium for a final volume of 5.0 mL and subsequently cultivated under the same conditions until an OD 600 nm of 0.3 was reached. At this point, arabinose was added to the medium to a final concentration of 0.05%, and the cultures were maintained at 30 °C and 200 rpm. After a minimum of 2 h of induction, 5 µL drops of cell cultures were placed on agarose-covered microscope slides for direct microscope observation [21]. For chromosome visualization, *X. citri* wt, Δ*envC* and Δ*envC* pMAJIIc-*envC* cells were cultivated under the same conditions described above and stained with DAPI using the protocol described in Ref [22]. Bacterial visualization was conducted using an Olympus BX61 microscope equipped with a monochromatic OrcaFlash2.8 camera (Hamamatsu, Japan) and TxRed and DAPI filters. Data collection and analysis were carried out using the CellSens Version 11 software (Olympus). Statistical analyses were conducted using GraphPad Prism version 6.

### 2.5. Pathogenicity and Bacterial Viability Analyses 

Pathogenicity tests were conducted in triplicate using Rangpur lime (*Citrus limonia*) as the plant host. Bacterial cultures (*X. citri wt*, Δ*envC* and Δ*envC* pMAJIIc-*envC*) were adjusted to 10^8^ CFU/mL (OD 600 nm of 0.3) using sterile 0.9% NaCl solution and then inoculated on the abaxial surface of leaves using a needleless syringe. The negative control consisted of inoculation with sterile 0.9% NaCl solution. The inoculated plants were kept in a controlled environmental plant laboratory equipped with a HEPA filter to maintain air particle purity. The conditions were set at 28–30 °C, 55% humidity and a 12 h light cycle, and the plants were observed for up to 30 days to monitor the appearance of citrus canker symptoms. The inoculated plants were assessed at 3, 5, 7, 10, 12 and 15 days after inoculation (DAI).

### 2.6. Growth Curves

*X. citri* wt, Δ*envC* and Δ*envC* pMAJIIc-*envC* were initially cultivated in NB medium for 16 h at 30 °C and 200 rpm. For the in vitro growth curves, the cultures were subsequently diluted in fresh NB medium to an OD 600 nm of 0.1 in a final volume of 1.5 mL (OD 600 nm of 0.3 corresponds to 10^8^ CFU/mL). Cell cultures were then distributed in the wells of a 24-well microtiter plate and incubated in a microtiter plate reader (Synergy H1N1; BioTek, Winooski, VT, USA) at 30 °C with constant agitation (200 rpm), and the OD at 600 nm was measured every 30 min for 72 h [23]. 

### 2.7. Phylogenetic Analyses and Protein Modeling

A single gene alignment—focusing on the XAC0024 gene sequence and sequences obtained from various *Xanthomonas* species (Appendix A)—was carried out using ClustalW [24]. For probabilistic analyses, the best evolutionary model was determined using jModelTest performed on the CIPRES resource [25]. Maximum likelihood (ML) analyses were conducted on RAxML version 8.0.24 [26], and branch support was assessed using bootstrap analysis [27] with 1000 replicates. The cladogram was drawn using MEGA X software version 10.1.5 [28].

## 3. Results

### 3.1. X. citri Encodes an EnvC Homolog, Which Is Conserved in MANY Bacteria

The XAC0024 protein exhibits a high degree of conservation among other sequenced *Xanthomonas* species, as shown in the maximum likelihood tree (Figure 1). Despite the fact that the single gene employed in the reconstruction may not precisely reflect the established phylogeny of the *Xanthomonas* genus, which is typically based on core genome alignment [29], our analysis successfully recovered some anticipated clusters. The species *X*. *citri* and *X*. *fuscans* demonstrate a close relationship and—in conjunction with *X*. *axonopodis* and *X*. *euvesicatoria* (*X*. *campestris* 85-10)—comprise the clade “*X*. *axonopodis*” [30]. This clade, incorporating the species *X*. *arboricola*, *X*. *fragariae*, *X*. *hortorum* and *X*. *gardneri*, aligns consistently with the topology of clade A, as reported in Ref [29], wherein *X*. *hortorum* and *X*. *gardneri* form a cohesive cluster [31].

A comparison of the nucleotide sequence between XAC0024 from *X. citri* 306 and its homolog XCC0022 from *X. campestris* pv. *campestris* str. ATCC 33913 showed an identity of 85% (Appendix A). Analysis of the protein sequences revealed that XAC0024 and XCC0022 shared an identity of 89%, a coverage of 98% and a similarity of 93% (Appendix A). These findings indicate a considerable level of sequence conservation between the two proteins. However, the comparison between the EnvC protein from *E. coli* and XAC0024 from *X. citri* revealed a 33% identity and a 52% similarity in terms of amino acid sequence (Appendix A). The EnvC protein from *E. coli* contains 16 additional amino acids at the N-terminus compared to its homolog XAC0024 from *X. citri.* To explore the shared conserved domains among these proteins, we utilized the NCBI Batch Web CD-Search tool. Figure 2 displays a visualization of the identified conserved domains between the three protein sequences from *X. citri* 306 (XAC0024), *X. campestris* (XCC0022) and *E. coli* (EnvC) (see Appendix A for a detailed visualization of the nine ORFs of *X. citri* sharing the M23 domain). 

To reinforce the similarity between the sequences of *Xanthomonas* strains used for phylogeny reconstruction, we utilized the NCBI Batch Web CD-Search tool, which revealed that the conserved M23 peptidase domain is shared by 16 representative *Xanthomonas* species (Appendix A). This observation suggests a potential conserved function of this domain among homologs in other bacteria [32]. An amino acid sequence analysis performed by the SignalP 5.0 server [33] revealed that EnvC from *X*. *citri*, *E*. *coli* and *X*. *campestris* possesses a signal peptide with specific cleavage sites (Appendix A). The cleavage site for EnvC in *X*. *citri* is between amino acids 20 and 21 (Appendix A), while for *E*. *coli*, it is between amino acids 42 and 43, and for *X*. *campestris*, it is between amino acids 14 and 15 (Appendix A).

### 3.2. Disruption of envC Affects X. citri Virulence 

To investigate the role of *envC*, we generated an *envC* deletion mutant (Δ*envC*) and assessed its virulence and viability in comparison with the wild-type isolate *X. citri* 306. These assessments included examining symptom development in planta and conducting in vitro growth curves. To determine the contribution of EnvC to virulence, Rangpur lime leaves were inoculated with *X. citri* wt, Δ*envC* and Δ*envC* complemented strain (Δ*envC* pMAJIIc-*envC*) and monitored for 15 days to observe the onset and progression of citrus canker symptoms. The temporal progression of symptoms showed hypertrophy/hyperplasia followed by water soaking and formation of brownish necrotic lesions at the late stage of infection, characteristic of citrus canker disease, in both *X. citri* wt and Δ*envC* pMAJIIc-*envC* strains (Figure 3A). However, the Δ*envC* mutant displayed a delay in the induction of citrus canker symptoms and produced lesions of reduced severity. These appeared to be concentrated close to the point of inoculation (Figure 3A).

Furthermore, we examined whether *envC* is essential for cell viability and proliferation by monitoring its growth in NB medium. Our results demonstrated that Δ*envC* exhibited a distinct growth pattern compared to *X. citri* wt (Figure 3B). At the outset, the mutant exhibited an accelerated growth rate; however, Δ*envC* eventually reached a plateau with a significantly lower population compared to both *X. citri* wt and Δ*envC* pMAJIIc-*envC* strains. When inoculated into leaves of Rangpur lime (*Citrus limonia*), Δ*envC* achieved a population 500 times less than that of the *X. citri* wt at 10 days after inoculation (results not shown). These findings underscore the necessity of *envC* for the virulence of *X. citri* and its ability to colonize the host. Additionally, disruption of *envC* adversely affects cell viability and proliferation.

### 3.3. EnvC Is Required for X. citri Daughter-Cell Separation 

To determine the sub-cellular localization of EnvC encoded by *X. citri*, we expressed a version of the protein as an mCherry fusion (EnvC-mCherry) (Figure 4). *X. citri* EnvC-mCherry expressing cells (pMAJIIc-*envC*) exhibited a strong fluorescence signal, primarily concentrated around the edges of the cells, while the cytoplasm remained non-fluorescent (Figure 4B,C). This phenotype suggests that *X. citri* EnvC is predominantly localized in the periplasmic region of the cells. Importantly, the wild-type *X. citri* strain used as a control exhibited no detectable fluorescence emission (Figure 4E,F).

Next, the possible roles of EnvC in cell division and chromosome segregation were investigated by examining DAPI-stained cells of *X. citri* wild-type, Δ*envC* and Δ*envC* pMAJIIc-*envC* strains (Figure 5). The *X. citri* Δ*envC* cells exhibited abnormally shaped rods, somewhat curved (Figure 5D–F). Many cells were arranged in long-chained structures, displaying clear division constrictions, which occasionally gave rise to minicells (Figure 5G–I; indicated by arrows). Although Δ*envC* mutants appeared competent in the initial stages of the division process, they exhibited noticeable and detectable defects in late-stage division/separation processes.

For cells, which were not part of a chain, thus displaying an overall normal shape, we observed a significant difference in their average cell length compared to *X. citri* wild-type and Δ*envC* pMAJIIc-*envC* strains (Table 1). To quantitatively evaluate this difference, we measured 400 individual cells from each culture: *X. citri* wild type, Δ*envC* and Δ*envC* pMAJIIc-*envC*. *X. citri* wild type had an average cell length of 1.18 ± 0.22 µm, while Δ*envC* pMAJIIc-*envC* had an average cell length of 1.22 ± 0.29 µm. In contrast, *X. citri* Δ*envC* mutants exhibited an average cell length of 1.89 ± 0.38 µm. Additionally, we scored the percentages of observed abnormalities for *X. citri* Δ*envC* mutants (Table 1), with filamented chains comprising approximately 55.5% of the cells, while minicells accounted for 5.25% (*n* = 400).

Chromosome organization was visualized using 4′,6-diamidino-2-phenylindole (DAPI) staining (Figure 5). Cultures of the Δ*envC* mutant displayed a continuous distribution of chromosomal mass spanning across the elongated cells (Figure 5E,F,H,I). This continuous distribution in some cells possibly hindered septal closure. In contrast, both *X. citri* and Δ*envC* pMAJIIc-*envC* strains displayed a bilobed chromosome organization pattern with the expected normal distribution, indicating successful complementation of the mutant (Figure 5B,C,K,L). In all strains, a strong nucleoid accumulation signal was also noticeable in the middle or pole of the cells (Figure 5B,H,K), consistent with previous observations in the *E. coli* wt strain [34] and *X. citri* 306 wt strain [17].

## 4. Discussion 

The XAC0024 protein from *X*. *citri* isolate 306 [15] exhibits significant homology with both EnvC proteins from *E*. *coli* and EnvC (XCC0022) from *X*. *campestris*. This homology was confirmed through amino acid sequence comparison analyses. Notably, the proteins from *X. citri* and *X*. *campestris* share a remarkable 93% similarity in amino acid sequence, while the *X*. *citri* protein demonstrates a 52% similarity with the EnvC protein from *E*. *coli*. Furthermore, a previous study reported homology between the XCC0022 protein from *X*. *campestris* and the *E*. *coli* EnvC protein [13].

The EnvC protein encoded by *X. citri* has a predicted M23 peptidase domain, which is part of a superfamily of metallopeptidase, characterized by the presence of zinc in its active enzyme site [35]. This enzyme family includes the Nlpd from *E. coli*, a LytM factor, as well as EnvC, responsible for activating N-acetylmuramoyl-L-alanine amidases (AmiA, AmiB and AmiC). These amidases play a crucial role in daughter-cell separation, and the inactivation of genes encoding them results in long cell chain formation [7,8,36]. 

In this study, we demonstrated that the EnvC protein of *X. citri* was predominantly localized in the periplasmic region of the cells and that the *envC* mutant of *X. citri* (Δ*envC*) displayed a defect in cell separation, accompanied by distinct changes in cell morphology. Cultures of the Δ*envC* mutant exhibited elongated rods, long-chained cells and minicells, which indicated impaired daughter-cell separation. These findings are consistent with the results previously reported for *X. campestris* strains lacking *nlpD*, *envC* or *amiC1* [13], which also exhibited severe defects in daughter-cell separation. The physiological roles of peptidoglycan hydrolases, including EnvC, are still not fully understood, as the loss of an individual enzyme has little effect on growth and division, suggesting a functional overlap between numerous hydrolases [2]. To shed light on the specific role of EnvC, a collection of *E. coli* mutants lacking individual LytM factors (EnvC, NlpD, YgeR and YebA), as well as all possible combinations of them, were previously analyzed [7]. Among these mutants, only those with *envC* deletion failed to normally separate, further confirming the fundamental role of EnvC in proper cell separation. 

Our observations of chromosome segregation errors in the *X. citri* Δ*envC* mutant suggest that they are linked to the delayed division induced by the absence of EnvC. Interestingly, complementation of the mutant with *X. citri* Δ*envC* pMAJIIc-*envC* restored a normal phenotype, with no detectable morphological discrepancies compared to the wild-type strain. Although further investigation is warranted, these results indicate that the Δ*envC* mutant indeed experiences a late-division/separation defect, possibly leading to longer cell compartments and allowing the chromosomal mass to span across.

The *X. citri* mutant lacking *envC* exhibited a delay in citrus canker symptomatology compared with the wt and complemented strains. This delay in symptom development demonstrated that the deletion of EnvC had a significant impact on the bacteria’s virulence, as evidenced by its reduced ability to induce symptoms when inoculated into citrus leaves, which are susceptible hosts. The complemented strain Δ*envC* pMAJIIc-*envC* fully restored virulence, confirming that the phenotype observed in the Δ*envC* mutant was specifically caused by the absence of EnvC and was not a result of polar mutation. 

Interestingly, *Xanthomonas campestris* strains lacking either *nlpD* or *amiC1* almost completely lost their virulence, but the mutant lacking *envC* showed levels of virulence comparable to those of the wt strain [13]. This indicates that the contribution of EnvC to virulence is species-specific and that EnvC may be directly or indirectly involved in specific virulence pathways, which differ between *Xanthomonas* strains. The species-specific role of EnvC in virulence highlights the complex and intricate nature of bacterial pathogenesis, where different strains of the same genus may employ distinct mechanisms to establish disease. Further investigations into the specific pathways influenced by EnvC in different *Xanthomonas* strains could provide valuable insights into the molecular basis of pathogenicity and potential targets for disease control strategies.

The presence of a signal peptide in EnvC from *X. citri*, *E. coli* and *X. campestris*, which can serve as a membrane anchor, suggests that EnvC proteins are likely to be transported to the periplasmic region, a common localization for proteins in many pathogenic bacteria, often facilitated by tat/sec systems in Gram-negative bacteria [37,38,39,40,41,42,43,44,45,46]. The localization patterns observed here for EnvC-mCherry fusion protein in *Xanthomonas citri* and *E. coli* [7] are consistent with the fusion protein occupying the periplasmic region of the bacterium [17]. This sub-cellular localization supports the notion that EnvC functions in the periplasm, where it may play a crucial role in late-division and daughter-cell separation processes, as supported by similar findings in other bacterial models [6,7,12,13].

Taken together, our results—as well as similar results from many other bacterial models—support the notion that EnvC has an important function in late-division and daughter-cell separation processes [6,7,12,47]. However, the mechanism by which *X. citri* EnvC operates in daughter-cell separation, considering its periplasmic location, and how and whether this protein interacts with other hydrolases are still intriguing questions, which require further investigation.

## 5. Conclusions

The present study demonstrates the critical role of the EnvC protein in *X. citri* virulence, as disruption of the *envC* gene significantly reduces the bacterium’s ability to cause citrus canker. This underscores the importance of EnvC in disease progression. Additionally, EnvC is shown to be indispensable for proper cell division, as its absence results in morphological alterations and defects in daughter-cell separation, indicating its involvement in division processes. The comparison with *X. campestris* strains lacking *envC* suggests that the role of EnvC in virulence may vary among different species within the *Xanthomonas* genus. This highlights the intricate nature of bacterial pathogenesis and underscores the necessity for species-specific investigations.

## Figures and Tables

**Figure 1 microorganisms-12-00691-f001:**
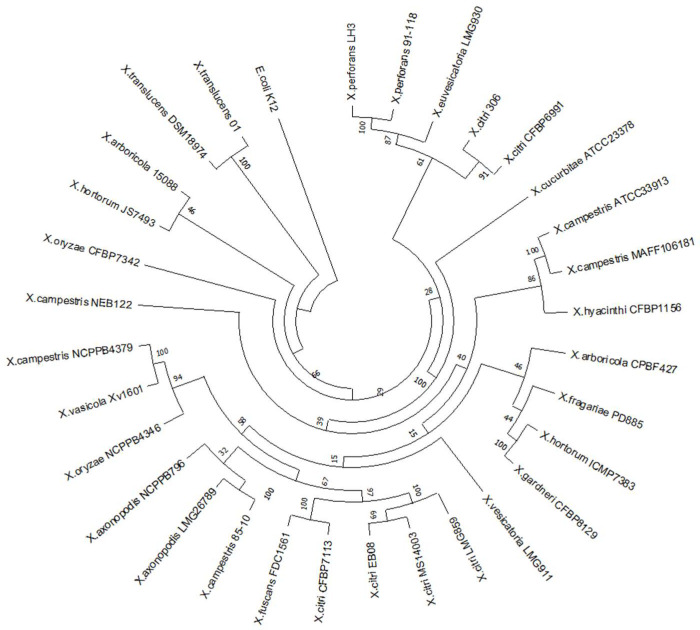
Maximum likelihood phylogeny based on the nucleotide sequences of 31 strains representing the *Xanthomonas* genus. The phylogenetic tree was inferred using RaxMLversion 8.0.24 and drawn using MEGA X software version 10.1.5. Values on the branches indicate bootstrap values for each branch, expressed as percentages. *E. coli* K12 was used as the outgroup.

**Figure 2 microorganisms-12-00691-f002:**
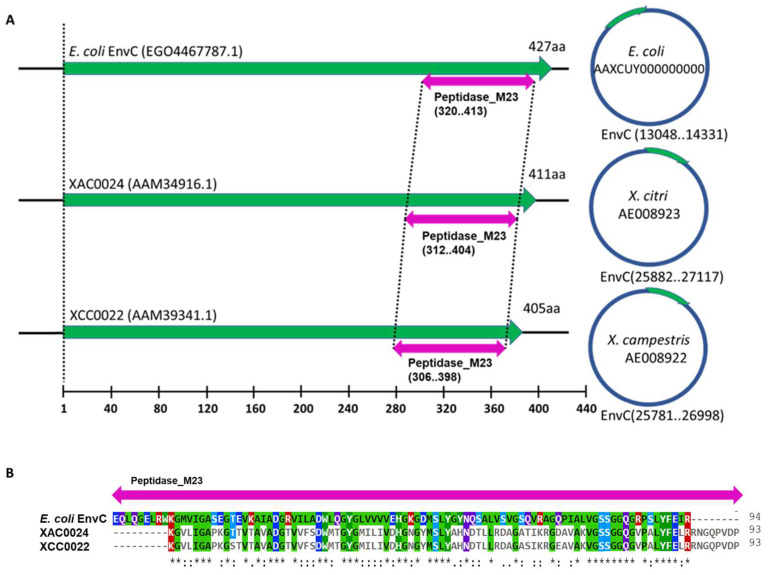
Multiple amino acid sequence alignment of M23 peptidase domain for XAC0024 from *X*. *citri* 306 (AAM34916.1), XCC0022 from *X. campestris* (AAM39341.1) and EnvC from *E*. *coli* (EGO4467787.1). (**A**) M23 peptidase domain within the protein sequences of *E. coli EnvC*, XAC0024 and XCC0022 (purple arrows) and their corresponding locus within genomes (circles on the right). (**B**) Multiple amino acid sequence alignment of M23 peptidase domain showing matching amino acids for the three organisms. The protein sequences were uploaded from the NCBI Batch Web CD-Search tool. Each protein is depicted with a Peptidase_M13 domain (pfam01551) and a conserved protein domain family EnvC (COG4942). * Conserved sequence (identical); : Conservative mutation; . Semi-conservative mutation.

**Figure 3 microorganisms-12-00691-f003:**
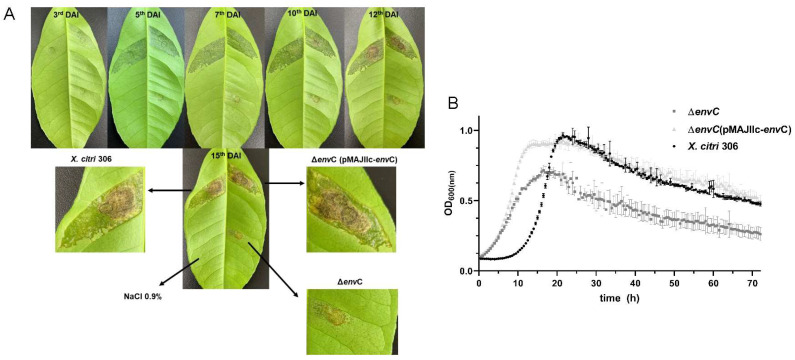
Pathogenicity assay and growth curve for *X. citri* 306, Δ*env*C and Δ*env*C-pMAJIIc-*env*C. (**A**) Rangpur lime leaves were infiltrated with cell suspensions of the specified *X. citri* strains. Pictures were taken at 3, 5, 7, 10, 12 and 15 days after inoculation (DAI). (**B**) In vitro growth curve. *X. citri* 306, Δ*env*C and Δ*env*C-pMAJIIc-*env*C were cultivated in NB medium, and OD 600 nm readings were taken every 30 min for 72 h. The points on the curves represent the average of triplicate cultures, and the vertical lines indicate the standard deviation values of the means. All experiments were performed in triplicate.

**Figure 4 microorganisms-12-00691-f004:**
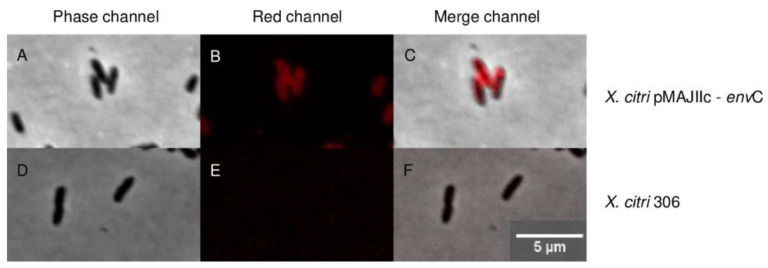
Sub-cellular localization of EnvC-mCherry in *X. citri*. *X. citri* strains expressing EnvC-mCherry fusions were cultivated up to OD 600 nm of 0.3, followed by induction with 0.05% arabinose for 2 h before microscope observation. The panels depict the phase contrast images (**left**), TxRed channels (**middle**) and the overlay, respectively, for (**A**–**C**): *X. citri* pMAJIIc-*env*C, (**D**–**F**): *X. citri* 306. Magnification 100X; scale bar 5 µm.

**Figure 5 microorganisms-12-00691-f005:**
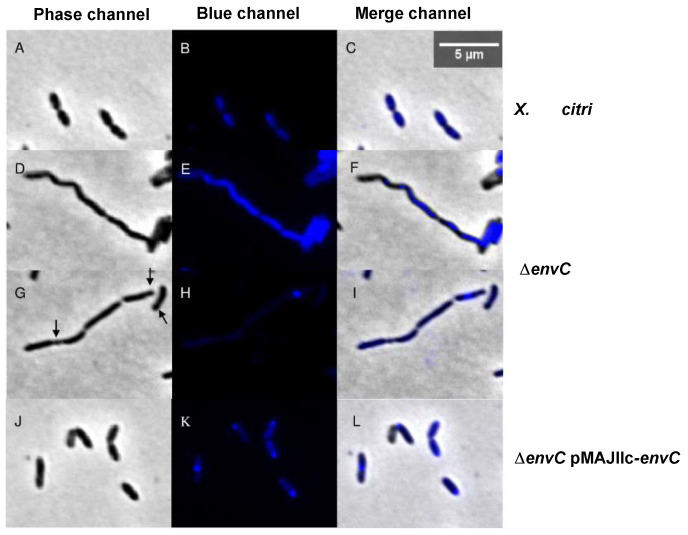
Cell morphology and nucleoid distribution analyses of *X. citri* 306, Δ*env*C and Δ*env*C pMAJIIc-*env*C strains. The panels show the phase contrast (**left**), DAPI channels (**middle**) and the overlay, respectively, for (**A**–**C**): *X. citri* 306, (**D**–**I**): Δ*env*C, (**H**–**L**): Δ*env*C-pMAJIIc-*env*C. Arrows indicate the minicell position. Magnification 100×; scale bar 5 µm.

**Table 1 microorganisms-12-00691-t001:** Morphological analysis of *X. citri* strains according to cell length.

	Cell Length µm	Filaments %	Minicells %
*X. citri* 306	1.18 ± 0.22 ^a^	0	0
Δ*envC*	1.89 ± 0.38 ^b^	55.5	5.25
Δ*envC* pMAJIIc-*envC*	1.22 ± 0.29 ^a^	0	0

Total *n* = 400 cells measured. Data correspond to the average cell length ± standard deviation. Same letters mean no significant difference according to the Tukey test—0.05.

## Data Availability

Data are contained within the article and Appendix A.

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
