# Peer review of "EnvC Homolog Encoded by Xanthomonas citri subsp. citri Is Necessary for Cell Division and Virulence"

_microorganisms, 2024, doi:10.3390/microorganisms12040691_

Round 1

Reviewer 1 Report

Comments and Suggestions for Authors

Row 29-30          - some suggestions hao to modify text: the role of EnvC in regulation of cell wall organization, cell division and clarifying the role of this peptidase in X. citri virulence.

Row 44 – missing dot end the end

Row 51 – E.coli  - in italic

Row 209 – better than photos were taken – inoculated plants were assessed

The methods are described in great detail and are adequate for the experiments used.

Row 269 -Figure 2 - this figure should be replaced with a clear and illustrative diagram

Some parts of Results (chapter 3.1) are combination of results and discussion - it should be styled better

Figure 5 – there are visible bacterial cells, but periplasmic localization?, it would be appropriate to provide a higher quality image with significantly better magnification

Row 417 – order od references – 7,8,38

the discussion is weak and should be improved, entire paragraphs rather describe results and are not a discussion, they do not contain references

it would be appropriate to summarize the results in the conclusion

inconsistent reference format

Reviewer 2 Report

Comments and Suggestions for Authors

The manuscript entitled “The EnvC homolog encoded by Xanthomonas citri subsp. citri is necessary for cell division and virulence” reported the role of EnvC in virulence and cell division. Overall, this research did not provide solid evidence to support the conclusion, especially the role for cell division.

1. The writing needs to be critically improved to meet publication requirement. There are too many grammar errors and typos throughout the manuscript.

2. table 1: use the formal or standard format suggested by the journal

3. figure 1:  calculation method was missed in figure caption

4. figure 2:  this is raw data from sequence Blast analyses, can not be used as a figure in main text.

5. figure 3: Comparison with 3-D structure of EnvC from X. citri and E. coli make no sense for the main findings in this manuscript.

6. figure 4: The growth of complemented strain was missed in B. In B and C, statistical analysis is need to confirm whether difference among wt and mutant at every time point. 

7. figure 5: western blot is needed to verify the existence of EnvC-mCherry fusion in periplasmic region

Examples for grammar errors and typos

line 78 E. coli could be deleted

line 85-86 rewrite the sentence

line 97 and 111: "overlap extension approach" and "The double-joint PCR" were revised as same 

line 131 "Xgal"

line 147 "Mutant obtention"

line 183 "X.citri"

line 148 rewrite

line 166 "EnvC" could be gene ID

line 307-309 mutant was "unable to replicate" why its population was increased? 

Comments on the Quality of English Language

Extensive editing of English language required
